# Kaolin–Polyvinyl Alcohol–Potato Starch Composite Films for Environmentally Friendly Packaging: Optimization and Characterization

Noshabah Tabassum [1,*], Uzaira Rafique [1], Maria Qayyum [1], Abdallah A. A. Mohammed [2], Saira Asif [3,4] and Awais Bokhari [4,5,6,*]

1 Department of Environmental Sciences, Fatima Jinnah Women University, The Mall, Rawalpindi 46000, Pakistan; provc@fjwu.edu.pk (U.R.); mariaqayyum1438@gmail.com (M.Q.)
2 Department of Chemistry, College of Science, King Saud University, Riyadh 11451, Saudi Arabia; aadam@ksu.edu.sa
3 Department of Botany, Pir Mehr Ali Shah (PMAS) Arid Agriculture University, Rawalpindi 46000, Pakistan; sairaasif@uaar.edu.pk
4 Sustainable Process Integration Laboratory—NETME Centre, Faculty of Mechanical Engineering, Brno University of Technology—Technická 2896/2, 616 69 Brno, Czech Republic
5 Department of Chemical Engineering, COMSATS University Islamabad, Lahore Campus, Lahore 54000, Pakistan
6 School of Engineering, Lebanese American University, Byblos 4504, Lebanon
* Correspondence: noshabatabassum@fjwu.edu.pk (N.T.); awaisbokhari@cuilahore.edu.pk or bokhari@fme.vutbr.cz (A.B.)

**Abstract:** This research paper introduces an innovative methodology to produce biodegradable composite films by combining kaolin, polyvinyl alcohol (PVA), and potato starch (PS) using a solvent casting technique. The novelty of this study resides in the identification and implementation of optimal synthesis conditions, which were achieved by utilizing the Response Surface Methodology—Central Composite Design. The study defines starch, polyvinyl alcohol (PVA), and kaolin as independent variables and examines their influence on important mechanical qualities, water absorption capacity, moisture content, and degradability as primary outcomes. The study establishes the ideal parameters as 5.5 weight percent Kaolin, 2.5 g of starch, and 3.5 g of PVA. These settings yield notable outcomes, including a tensile strength of 26.5 MPa, an elongation at break of 96%, a water absorption capacity of 21%, a moisture content of 3%, and a remarkable degradability of 48%. The study emphasizes that the augmentation of kaolin content has a substantial impact on many properties, including degradability, tensile strength, and elongation at break. Simultaneously, it leads to a reduction in the water absorption capacity and moisture content. The study's novelty is reinforced by conducting an additional examination on the ideal composite film, which includes investigations using FTIR, TGA, and SEM-EDX techniques. The consistency between the predicted and experimental results is noteworthy, as it provides further validation for the prediction accuracy of Design Expert software's quadratic equations. These equations effectively capture the complex interactions that exist between process parameters and selected responses. This study presents novel opportunities for the extensive utilization of PVA/PS composite films, including kaolin in various packaging scenarios, thereby significantly advancing sustainable packaging alternatives. The statistical analysis provides strong evidence supporting the relevance of the models, hence increasing our level of trust in the software's prediction skills. This conclusion is based on a 95% confidence level and $p$-values that are below a threshold of 0.05.

**Keywords:** biodegradable composites; kaolin; polyvinyl alcohol; potato starch; response surface methodology

## 1. Introduction

The last 20 years indicate an enormous increase in the production and utilization of plastics worldwide, leading to an increase in waste disposal problems. According to Agarwal [1], the plastic sector is one of the largest revenue-generating sectors globally, with a market size of USD 348 billion. It is estimated that there will be a continuous annual increase of 4.2% from 2021 to 2026. The packaging sector, encompassing items such as plastic bags, plastic bottles, and product wrappings, contributes to 30% of the total plastic consumption. Similarly, the building and construction sector, involving plastic pipes and vinyl coverings, accounts for 17%, while the transportation sector, including glazing, interior wall panels, partitions, and headliners, constitutes 14% [2]. The most used synthetic plastics across these sectors are low- and high-density polyethylene (LDPE, HDPE), polypropylene (PP), polyvinyl chloride (PVC), polystyrene (PS), and polyethylene terephthalate (PET). Collectively, these plastics make up approximately 90% of the total plastic production [3]. The major raw materials used for packaging applications are polyethylene and polypropylene, which contribute significantly to the country's global economy [4]. Despite their low cost and good mechanical and thermal properties, these materials are synthesized from crude oil byproducts, making them the main culprit for global warming [5]. In addition to this, plastic pollution is causing the deaths of aquatic animals, the loss of habitats, and the pollution of landfills. According to previous studies, the arctic ice, once considered a pure and virgin environment, is now found to be contaminated with microplastics, with up to 10 thousand particles per liter of snow observed [6,7]. According to Lau et al. [8], if the current rate of plastic usage continues, it is projected that by 2050, 12 billion tonnes of plastic will accumulate in landfills and the environment. The consequence of this would be a depletion of 20% of our world's natural resources.

The need for a sustainable and environmentally friendly packaging solution has led to the substitution of synthetic petroleum-based polymers with natural polymers. Scientists and researchers are concerned with diminishing the adverse human impact on the environment, and bio-based composites are a major alternative. Biodegradable polymer composites have gained significant attention because of their exceptional mechanical strength, degradation potential, and biocompatible nature [9,10].

Starch is a ubiquitous, renewable, agriculture-based biopolymer. It is inexpensive and one of the most abundant polysaccharide sources frequently used to develop biodegradable packaging materials due to its unique biodegradable and biocompatible nature [11,12]. Starch comprises linear amylose and highly branched amylopectin and can form cohesive sheets under specific conditions, making it insoluble in cold water and incapable of melting like common thermoplastics. However, starch granules lose their semi-crystalline structure and become a matrix when subjected to shear forces, thermal energy, and softeners. The OH functional groups in starch can be modified through esterification and oxidation to produce biodegradable composite films from starch [13]. Despite its cost-effectiveness and suitable physical and mechanical properties, starch has a lower mechanical strength than synthetic polymers [9]. Starch films can be modified by adding other polymers like PVA, glycerol, sorbitol, and polyethylene glycol. Starch film's flexibility can be improved by adding glycerol. Researchers have extensively studied blends of starches and synthetic polymers since the time of superfluous advancement in the field of polymer science [14,15].

Various research efforts have also been made towards the development of PVA (polyvinyl alcohol)/starch blends. PVA is one of the most non-toxic, biodegradable, and semi-crystalline synthetic polymers that has good strength, film-forming ability, high resistance to oil and solvents, a gas barrier, and mechanical properties [16]. It is most suitable for blending with starch due to its polarity and synergistic interactions with the physical bonds found in both. PVA holds a large amount of free hydroxyl groups, which permits it to react with many other functional groups [17,18]. Crosslinking is the most inclusive approach to ameliorating the performance of PVA for various applications, and many studies have reported on various crosslinkers for PVA, including mineral clays, boric acid, weak Lewis's acid, aluminum chloride, titanium chloride, glyoxal, formaldehyde, zirco-

nium chloride, glutaraldehyde, etc. [19]. Most studies have preeminently focused on 2:1 layered clay-based composites, including Montmorillonite, Beidellite, Saponite, Hectorite, and Sauconite, due to their favorable physicochemical properties and their ability to be exfoliated [19]. However, 1:1-layered, less expandable clays like kaolin are seldom utilized.

Kaolinite, also referred to as Kaolin clay, is a naturally occurring clay mineral that is widely distributed across the globe, particularly in Asia, Europe, and Africa. It has diverse applications in adhesives, paints, pharmaceuticals, fiberglass, paper, rubber, ceramics, electronics, plastics, coatings, and agriculture. Kaolinite has a plastic nature and comprises a hydrated aluminum silicate, $Al_2Si_2O_5(OH)_4$, which has a structure of 1:1 dioctahedral asymmetric layers that are linked through apical oxygen. One side of the layer is siloxane and comprises silicate sheets ($Si_2O_5$) that are tetrahedrally joined to oxygen. The other side of the layer is gibbsite-like and consists of aluminum oxide/hydroxide ($Al_2(OH)_4$) that is octahedrally coordinated to oxygen and hydroxyls. Essentially, the mineral clay has a layer composed of one octahedral sheet that is consolidated with one tetrahedral sheet [20–23].

Kaolinite has been extensively studied by researchers because of its stable physico-chemical properties and cost-effectiveness. According to Mbey et al. [24], the addition of kaolin clay to starch films improved their thermal stability and acted as a barrier to UV light. These findings suggest that kaolinite is one of the most appropriate natural clays to be used in starch films. This versatile material has a wide surface area and provides a reactive surface for OH groups. As a result, the incorporation of naturally occurring mineral clays into packaging is becoming an interesting and cost-effective alternative.

Presently, there is no report about the inclusion of kaolin as a reinforcing agent in PVA/starch-based composite films. The reinforcement of a polymer matrix with mineral clays can further improve the adhesion, thermal, and mechanical properties of these materials. Therefore, the objective of the present study is to synthesize bio-composite films using starch, PVA, and kaolin, which can be utilized in the packaging industry, and to investigate the impact of kaolin on PVA–starch films. Numerous factors can influence the physicochemical properties of synthesized films, either individually or synergistically. They can be optimized through various experimental design techniques [25,26].

In this study, the response surface methodology (RSM) was utilized to optimize the variables that impact the characteristics of starch-based films. The RSM is a statistical technique employed to model and optimize complex systems. By conducting a series of experiments and analyzing the data using the RSM, we determined the optimal conditions for producing the composite films and assessed the influence of variables on their properties. This systematic approach facilitated an efficient investigation of the formulation and characterization processes of composite films. The starch, polyvinyl alcohol, and kaolin ratios were taken as variables that affect the film's overall properties in the experimental design. The synthesized composite material based on PVA, potato starch, and kaolin offers improved mechanical strength, enhanced barrier properties, reduced moisture sensitivity, and is environmentally suitable. It outperforms conventional packaging materials by combining the strength of kaolin with the biodegradability of PVA and potato starch. This innovative approach aligns with the increasing demand for sustainable packaging solutions.

## 2. Material and Methods

### 2.1. Materials

Starch was extracted from *Kuroda* red skin potatoes. The extraction procedure of Arikan and Bilgen [27] was adopted with slight modifications. Polyvinyl alcohol (molecular weight: 85,000–124,000; hydrolyzed 99+%; viscosity: 28.0–32.0 cps), glycerol (molecular weight: 92.09; purity: ≥99.0%), and kaolin (particle size: ≤1 μm; surface area: 20.5 $m^2$/g; free moisture content: 1%; pH: 6–8; specific gravity: 2.6) were purchased from Sigma Aldrich (Merck KGaA, Darmstadt, Germany).

## 2.2. Experimental Design

Statistical analysis and process optimization were applied via Design Expert 8.0.7.1 software (Stat-Ease, Inc., Minneapolis, MN, USA). Starch, PVA, and kaolin were optimized by employing RSM-CCD). The responses evaluated were water absorption, moisture content, degradability, and mechanical properties. The coded values and their levels for CCD are given in Table 1. The complete design package comprised 17 combinations with three replicates at a central point to determine the relationships between the process parameters and defined responses, and the experiments were executed randomly.

**Table 1.** Levels of independent variables for DOE (Design of Experiments).

| Symbols | Independent Variables | Minimum Level | Mean | Maximum Level |
|---------|----------------------|---------------|------|---------------|
| A | Starch (g) | 0 | 2.50 | 5.02 |
| B | PVA (g) | 0.98 | 3.50 | 6.02 |
| C | Kaolin (%) | 0 | 5.62 | 13.07 |

The optimal conditions for developing composite films were predicted by relating process parameters to each response via a quadratic equation, as given below.

$$Y = \beta_{\circ} + \sum_{i=0}^{n} + \beta_i \chi_i + \sum_{i=0}^{n} + \beta_{ii} \chi_i^2 + \sum_{i \neq i=1}^{n} + \beta_{ij} \chi_i \chi_j \qquad (1)$$

An ANOVA and three-dimensional graphs were used to indicate the interaction between process variables and responses. The goodness of fit of the polynomial model was assessed through the coefficient of determination ($R^2$).

## 2.3. Preparation of Films

The films were prepared via the solvent casting process based on the conditions presented in Table 2. The PVA/PS blends were prepared by dispersing them in 50 mL of distilled water at 70 °C with continuous electric stirring at 300 rpm using JOANLAB (OSC-10L) equipment. In the next step, Kaolin, as a reinforcing agent, was added in the concentrations specified in Table 2, along with glycerol (3% wt of the total solution), and the solution was continuously blended electrically for 120 min until the mixture became viscous. After cooling, the sample was cast onto a plexiglass sheet (2 mm, 8 × 12 inches) and leveled using a 20 μm casting knife. Subsequently, the films were subjected to solvent evaporation by placing them in a hot oven at 45 °C for 24 h. The thickness of each film was measured at five different points using a digital micrometer (0–25 mm, TERMA BRAND, Hangzhou, Zhejiang, China), and the average was calculated to confirm the uniformity of the fabricated films. The analyses were conducted under environmental conditions of 27 ± 2 °C temperature and 40 to 47% relative humidity. The primary focus was to evaluate the influence of kaolin on the PVA/PS matrix. The visual appearance of the films was not the sole criterion used to assess their quality. Rather, the physicochemical properties were primarily considered to determine the optimal conditions for producing the best films. To ensure accuracy and reliability, all experiments are as shown in Figure 1, and measurements were carried out with a minimum of three replicates.

**Table 2.** Experimental design parameters and measured responses.

| | | Parameters | | | Responses | | | | |
|-----|------|--------------|------------|--------------|----------------------------|-------------------|--------------------------|--------------------------|--------------------|
| Std | Expt | Starch (g) | PVA (g) | Kaolin (%) | Tensile Strength (Mpa) | Elongation (%) | Water Absorption (%) | Moisture Content (%) | Degradability (%) |
| 7 | 1 | 1.00 | 5.00 | 10.00 | 15 | 84 | 8 | 5 | 34 |
| 14 | 2 | 2.50 | 3.50 | 13.07 | 10 | 79 | 6 | 4 | 30 |

**Table 2.** *Cont.*

| | | Parameters | | | Responses | | | | |
|---|---|---|---|---|---|---|---|---|---|
| Std | Expt | Starch (g) | PVA (g) | Kaolin (%) | Tensile Strength (Mpa) | Elongation (%) | Water Absorption (%) | Moisture Content (%) | Degradability (%) |
| 6 | 3 | 4.00 | 2.00 | 10.00 | 14 | 80 | 9 | 4 | 29 |
| 10 | 4 | 5.02 | 3.50 | 5.50 | 17 | 84 | 14 | 6 | 26 |
| 1 | 5 | 1.00 | 2.00 | 1.00 | 7 | 38 | 32 | 20 | 25 |
| 2 | 6 | 4.00 | 2.00 | 1.00 | 9 | 50 | 38 | 28 | 16 |
| 16 | 7 | 2.50 | 3.50 | 5.50 | 20 | 89 | 15 | 3 | 39 |
| 5 | 8 | 1.00 | 2.00 | 10.00 | 10 | 75 | 10 | 5 | 29 |
| 12 | 9 | 2.50 | 6.02 | 5.50 | 16 | 89 | 20 | 8 | 36 |
| 11 | 10 | 2.50 | 0.98 | 5.50 | 9.9 | 50 | 14 | 6 | 29 |
| 4 | 11 | 4.00 | 5.00 | 1.00 | 11 | 76 | 45 | 32 | 18 |
| 15 | 12 | 2.50 | 3.50 | 5.50 | 19 | 88 | 18 | 4 | 38 |
| 3 | 13 | 1.00 | 5.00 | 1.00 | 9.8 | 74 | 44 | 40 | 17 |
| 13 | 14 | 2.50 | 3.50 | 0.00 | 13 | 75 | 51 | 33 | 19 |
| 9 | 15 | 0.00 | 3.50 | 5.50 | 10.8 | 70 | 12 | 15 | 34 |
| 17 | 16 | 2.50 | 3.50 | 5.50 | 20.5 | 94 | 22 | 3 | 45 |
| 8 | 17 | 4.00 | 5.00 | 10.00 | 17 | 90 | 13 | 5 | 40 |

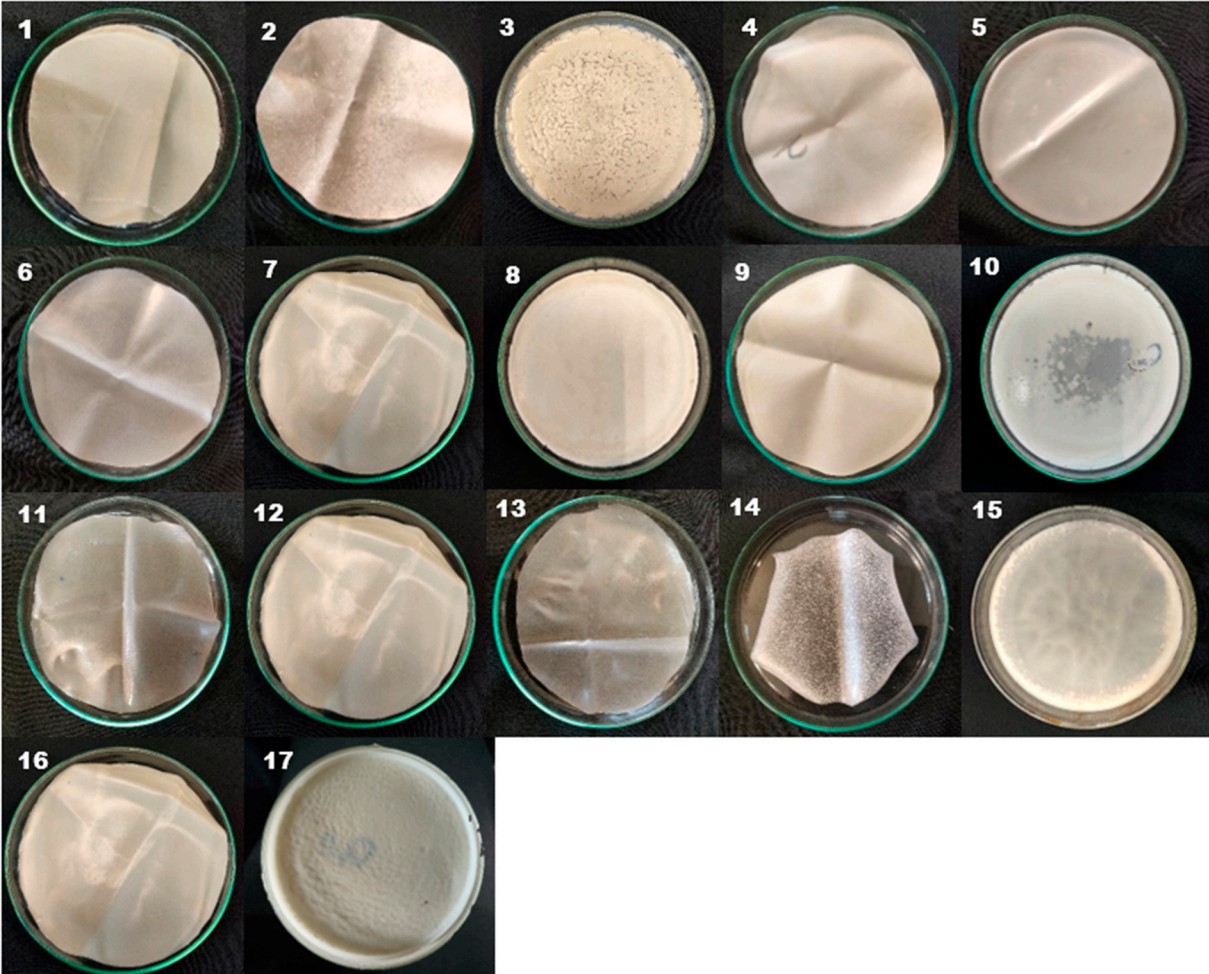

**Figure 1.** The prepared films with varying amounts of starch, PVA, and kaolin based on the experimental design.

### 3. Characterizations

*3.1. Thermal Properties*

The thermal properties of PVA, PS, kaolin, and optimal composites were characterized through TGA coupled with DSC using the SDT Q600 (TA Instrument, New Castle, DE, USA). The specimens, weighing between 10 and 25 mg, were subjected to a temperature increase of 10 °C/min, starting at room temperature and going up to 500 °C.

*3.2. FTIR Analysis*

A spectrophotometer (SHIMADZU 8300, Kyoto, Japan) was utilized to conduct Fourier transform infrared (FTIR) experiments at a resolution of 4 cm$^{-1}$ with triplet scans of kaolin, PVA/PS, and optimal composite.

*3.3. Morphological Analysis*

The SEM-EDX analysis of the composite films was conducted using a Scanning Electron Microscope (MIRA3 TESCAN, Brno, Czech Republic). The mounted film samples were affixed to a stub using double-sided adhesive tape and then covered with a thin layer of gold (10–20 nm). The images were recorded at a voltage rate of 5–10 kV and a resolution of 5 nm.

*3.4. Mechanical Analysis*

The tensile strength (TS, MPa) and elongation at break (EAB, %) of the films were measured by following ASTM D882-12 using the WAW-500E computerized hydraulic universal tensile testing machine [28,29]. This machine provides a maximum load of 500 kN with $\leq\pm1\%$ load and deformation accuracy. The machine has two important zones for testing films: the upper zone, which is fixed and provides tension, and the lower zone, which is movable for the compression, stretching, and bending of a sample. The machine's clamps are designed to accommodate both round and flat samples. However, in this study, flat clamps with a thickness range of 0–30 mm were used for the fabricated films. The machine offers a maximum distance of 750 mm between the two tensile grips. In accordance with the fabricated films of this study, the initial adjustment was set at 40 mm, and before testing, all films were cut into strips (100 mm × 40 mm) and left to stabilize at 24 °C and 50% relative humidity for three days [30]. The ends of the films were mounted vertically between the clamps, and the extensional speed was set to 0.2 kn/s. A photoelectric encoder, a sensing device installed in the machine, transmitted the movements of the samples outside as an electrical signal. Tensile data were collected using TE software, and to ensure accuracy, each experiment was repeated three times.

*3.5. Water Absorption*

The water absorption capacity is considered a crucial property for evaluating the performance of bio-based films in diverse applications. It deals with the ability of bioplastic films to take in and retain water under explicit conditions. The water potential of the films is affected by factors such as the presence of hydrophilic functional groups in the polymer structure, the degree of crystallinity, and the porosity of the film. In this investigation, the water absorption potential of the films composed of kaolin and the PVA/PS matrix was assessed in triplicate using a method described by Shen et al. [31] with minor adaptations. The hydrophilic nature of PVA/PS arises from the presence of hydroxyl (-OH) groups along their polymer chains. These OH groups readily interact with water molecules through H-bonding, facilitating the absorption of water into the matrix. In addition to that, understanding the chemistry of kaolin with respect to water absorption is crucial, especially when incorporating it into matrices. Kaolin possesses a porous and layered crystal structure with OH groups on its surface. When it interacts with PVA and PS, it enhances the overall adhesion, stability, and compatibility between the components, contributing to the improved water absorption properties of the film.

This test was performed by submerging dry 50 × 50 mm films in 100 mL of distilled water at room temperature for 24 h, and the non-soluble film was examined for any dimensional changes after gently dabbing the surface with a Whatman filter paper. The water absorption capacity was calculated according to the equation WA (%) = [(Ww − Wd)/Wd] × 100, where Ww is the weight of the wet film after sinking, and Wd is the weight of the dry film before sinking.

### 3.6. Moisture Content

The moisture content of the prepared composite films was evaluated using the following procedure. Film samples measuring 40 mm × 40 mm were cut and dried at 50 °C, and the initial mass ($M_1$) was recorded. The samples were then placed in a desiccator along with aqueous, saturated NaCl solutions [32], creating a 75% constant RH environment. After a full day, the film samples were weighed to determine their final mass ($M_2$). The moisture content of the films was calculated using the following equation: MC (%) = [($M_2$ − $M_1/M_1$)] × 100.

### 3.7. Soil Burial Degradability

Samples with a dimension of 6 × 3 $cm^2$ were buried into sandy loam soil at a 10 cm depth under temperatures of 32 °C and 6.5 and an environmental relative humidity (RH) of around 57% [33].

## 4. Results and Discussion

### 4.1. Statistical Analysis

The statistical analysis of the PVA/PS kaolin-based composite films was investigated in terms of an analysis of variance, *F*-value, *p*-value, $R^2$, adjusted $R^2$, predicted $R^2$, adequate precision, and coefficient of variance. (Tables 3 and 4).

**Table 3.** Analysis of variance (ANOVA) for tensile strength (TS-MPa), elongation at break (E%), water absorption capacity (WA%), moisture content (MC%), and degradability (D%) properties.

| | TS(MPa) | | E (%) | | WA (%) | | MC (%) | | D (%) | |
|---|---|---|---|---|---|---|---|---|---|---|
| **Source** | *F*-Value | *p*-Value | *F*-Value | *p*-Value | *F*-Value | *p*-Value | *F*-Value | *p*-Value | *F*-Value | *p*-Value |
| Model | 17.27 | 0.0005 | 17.19 | 0.0006 | 24.18 | 0.0002 | 24.99 | 0.0002 | 23.88 | 0.0002 |
| A | 17.76 | 0.0040 | 6.86 | 0.0345 | 1.32 | 0.0482 | 2.60 | 0.1511 | 3.77 | 0.0934 |
| B | 24.14 | 0.0015 | 63.42 | <0.0001 | 4.29 | 0.0772 | 3.92 | 0.0481 | 4.76 | 0.0454 |
| C | 25.48 | 0.0015 | 39.94 | 0.0004 | 207.35 | <0.0001 | 185.99 | <0.0001 | 118.39 | <0.0001 |
| AB | 0.96 | 0.0403 | 0.74 | 0.4178 | 0.21 | 0.0437 | 1.10 | 0.0298 | 6.64 | 0.0366 |
| AC | 0.32 | 0.0472 | 0.23 | 0.0487 | 8.383 | 0.037 | 0.11 | 0.0450 | 4.82 | 0.042 |
| BC | 0.33 | 0.5851 | 12.04 | 0.0104 | 1.25 | 0.0005 | 2.75 | 0.0411 | 10.41 | 0.0145 |
| $A^2$ | 26.09 | 0.0014 | 9.99 | 0.0159 | 2.20 | 0.1817 | 15.68 | 0.0055 | 25.54 | 0.0015 |
| $B^2$ | 25.81 | 0.0006 | 26.65 | 0.0013 | 4.458 | 0.0486 | 6.05 | 0.0434 | 14.40 | 0.0068 |
| $C^2$ | 77.53 | <0.0001 | 19.86 | 0.0029 | 12.78 | 0.0090 | 58.13 | 0.0001 | 93.41 | <0.0001 |

The *p*-values for all the models in Table 3 were less than 0.05, indicating that they were all significant at the 95% level of confidence. The variables and the process responses were significantly related as expected; the F-statistic was not significant. By means of comparing the predicted values to the actual values, the $R^2$ coefficient was calculated. Actual values are the experimental values obtained from performing experiments in the laboratory (Table 2), while predicted values are values anticipated by a statistical model based on the input of certain given variables, as shown in Figure 2. Following is the equation for the $R^2$ coefficient, SSR/SST, where SSR is the sum of squares due to regression and SST is the total

sum of squares. This statistical measure specifies how accurately the model fits the data. An $R^2$ value $\approx 1$ indicates a large proportion of variability in the data, whereas a lower $R^2$ value represents that the model does not truly fit the data. As can be seen in Table 4, the $R^2$ values for all the models are close to 1. This indicates a strong correlation between the process variables and the relevant responses.

**Table 4.** Fit statistics of the obtained model for each dependent variable: tensile strength (TS-MPa), elongation at break (E%), water absorption capacity (WA%), moisture content (MC%), and degradability (D%) properties.

|  | TS (MPa) | E (%) | WA (%) | MC (%) | D (%) |
|---|---|---|---|---|---|
| $R^2$ | 0.9569 | 0.9567 | 0.9688 | 0.9698 | 0.9685 |
| Adj $R^2$ | 0.9015 | 0.9010 | 0.9288 | 0.9310 | 0.9279 |
| Pred $R^2$ | 0.7187 | 0.7260 | 0.7830 | 0.7363 | 0.8252 |
| Adequate Precision | 13.384 | 13.588 | 14.038 | 14.086 | 14.279 |
| C.V% | 4.95 | 4.24 | 8.88 | 4.99 | 4.11 |

Additionally, it was noticed (Table 3) that the Adj $R^2$ values for all the models were in reasonable agreement with the Pred $R^2$. This indicates that the model is not underfitting or overfitting the data. Low Pred $R^2$ values occur when the model is too simple and does not capture the relationships in the data, and this resonates with underfitting activity. Whereas overfitting relates to the complexity of the model, noise capture in the data, and poor performance on new data, which can result in a high Pred $R^2$ value. On the contrary, when the Adj $R^2$ values are close to the Pred $R^2$ values, this indicates that the model is effectively capturing the correlations in the data without underfitting or overfitting and can be used to make reliable predictions on new data [33–37].

It can be seen in Table 4 that all the models have adequate precision (AP) values greater than four, which means that the model is accurate enough and can predict new values. This ensures that the model can be used to direct the design space specified by CCD.

The model is considered reproducible if its CV is less than 10%. It can be seen in Table 4 that all the predicted models have CV values < 10%, which indicates that all the models can be reproducible.

### 4.2. Effect on Mechanical Properties

The mechanical properties of the prepared samples were evaluated, and the results for tensile strength and elongation at break are illustrated in Table 2. The specimens exhibited various characteristics, such as toughness, brittleness, and flexibility. The highest tensile strength was observed in Expt 16, while the lowest values were recorded for Expt 5 and Expt 6. Moreover, the highest elongation at break was observed in Expt 7, Expt 9, and Expt 16, while the lowest values were recorded for Expt 5, Expt 6, and Expt 10. A composite film with satisfactory mechanical properties should possess both appropriate stress and strain [38]. Previous research has reported tensile strengths ranging from 3 to 20 MPa for films made from a polymer matrix and nanoparticles [39], while another study reported a range of 0.1 to 8.8 MPa for films made from semolina flour reinforced with different plasticizers [40]. Furthermore, Zhao et al. [41] fabricated silica-based composites with a high stress of 0.18 MPa and a strain of 60%. These properties uniquely advantage the composite aerogel in thermal system optimization.

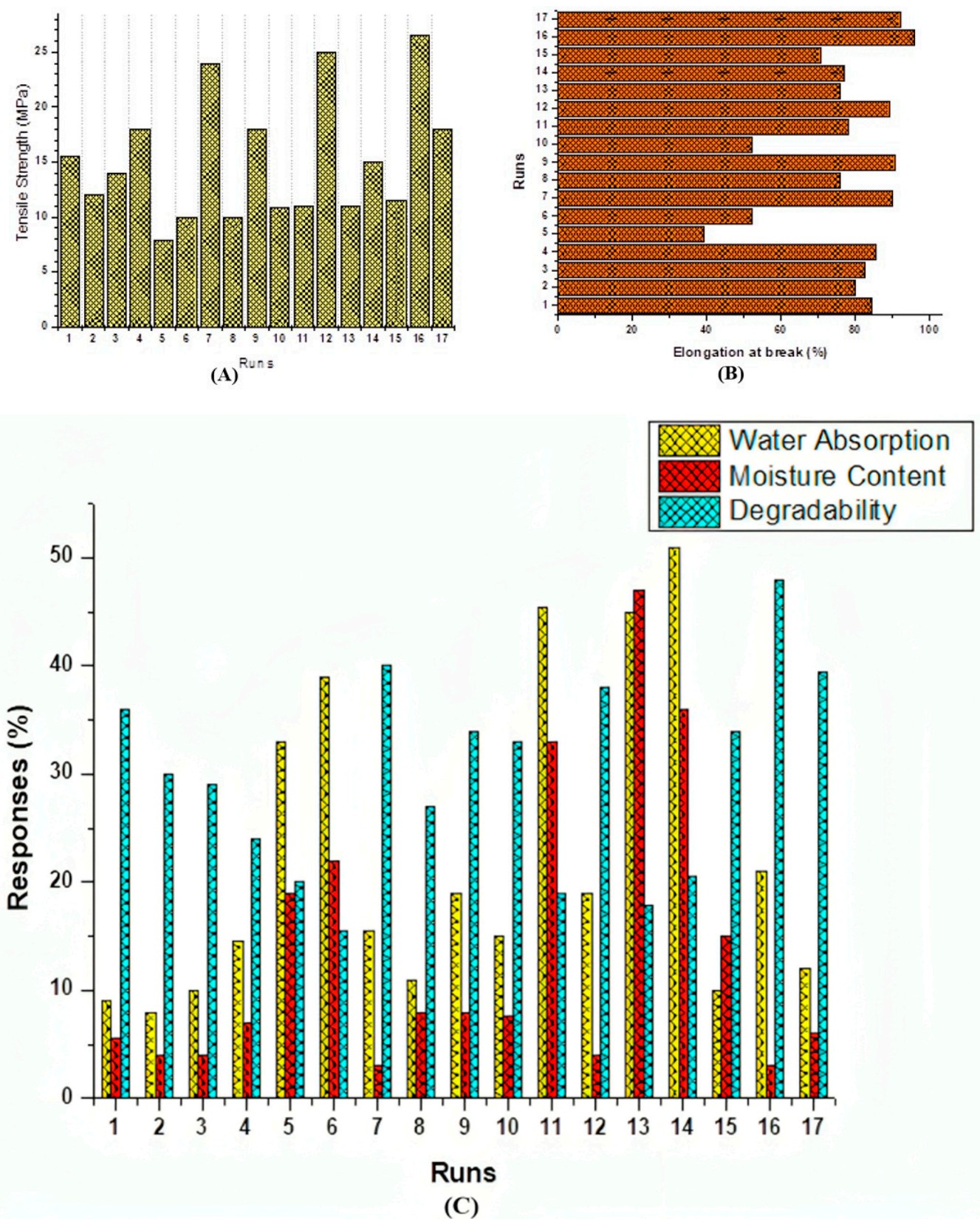

**Figure 2.** Graphs of predicted responses of physicochemical properties of prepared films: (**A**) tensile strength; (**B**) elongation at break; and (**C**) water absorption, moisture content, and degradability.

The findings of the study on the effect of kaolin on the tensile strength of PVA/PS composite films are presented in Figure 3, which displays three-dimensional response

surface plots. The plots revealed that the amount of kaolin on the X2 axis had a significant impact on the tensile strength of the composite films, as shown in Figure 3A. An increase in the amount of kaolin led to an increase in the film's strength, while a decrease in its amount resulted in a decrease in the strength of the film. This can be attributed to the load-bearing component in the alloy, which changes with the amount of kaolin. Additionally, the interaction between PVA and kaolin is crucial. It can be seen in Figure 3A,B that when the concentration of PVA fluctuated on the X1 axis, it had a significant impact on the tensile properties of the composite film. The findings in Table 2 reveal that in Expt 9, the PVA concentration was 6.02 g, while in Expt 11, it was 0.98 g. In the former, the composite exhibited a TS of 16 MPa with an 89% EAB, whereas in the latter, it showed a TS of 9.9 MPa with a 50% EAB. The lower concentration of PVA made the films more brittle, leading to a decrease in elongation. In this way, the same trend was observed for starch on the X1 axis in Figure 3A,B. In Expt 17, the starch concentration was 4 g, while in Experiment 15, it was 0 g. The first one showed a TS of 17 MPa with a 90% EAB, whereas the second one exhibited a TS of 10.8 MPa with a 70% EAB. Interestingly, with the same starch concentration of 4 g in Expt 3, 6, and 11, the tensile properties declined. This suggests that the distribution of PVA and PS in the composite film plays a significant role in its tensile properties. Optimal concentrations of PVA and PS are crucial to achieving the desired balance between tensile strength and elongation at break.

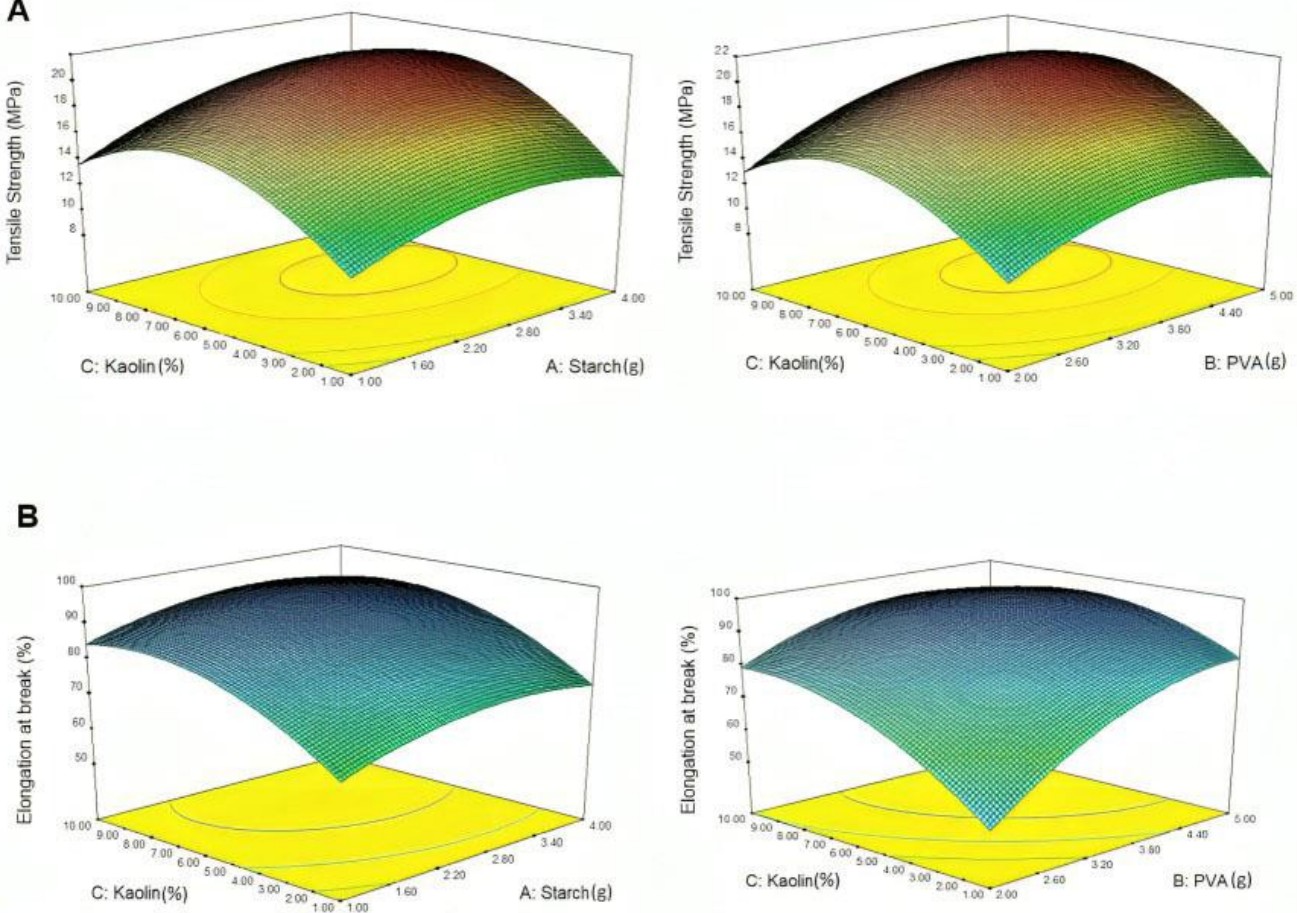

**Figure 3.** Three-dimensional response surface methodology (RSM) graphs (**A**) showing the impact of the interaction of kaolin with starch and PVA on the tensile strength (MPa) of the films (**B**) and on the elongation at break (%).

The dome-shaped graphs presented in Figure 3B indicate that the minimal or no concentration of kaolin resulted in a very limited elongation at break. However, with an increase in the concentration of kaolin from 0% to 13%, an average increase of 30% in

the elongation at break was observed in the PVA/PS composite matrix. This increase is attributed to the effective anchoring of the PVA starch-based composite onto the exfoliated clay plates, which is facilitated by the good dispersion of the clay. Additionally, it was observed that an increase in the amount of kaolin reduces the amount of starch, which in turn decreases the tensile strength of the film. However, this trend shows an optimal value for PVA of around 3.5 g. The concentration of glycerol as a plasticizing agent was kept constant in the matrix, and its three groups of hydroxyls enhanced the hydrogen interactions between the starch/PVA and kaolin chains, making glycerol an important factor in strengthening the matrix interface.

Conversely, the elongation at break of the Expt 2 composite film decreased as the kaolin loading increased. The film exhibited an elongation at break of 79% at a 13.0 wt% kaolin addition and 94% at a 5.5 wt% kaolin addition. This is due to the reinforcing effect of kaolin, which reduces the mobility of starch chains, resulting in increased rigidity and brittleness. These findings are consistent with the results reported by Shanmathy et al. [42], who observed a significant reduction in the elongation at break of starch films with increasing bentonite loading from 0 wt% to 2.5 wt%.

### 4.3. Effect on Water Absorption Capacity

In Figure 4, the effect of the kaolin content on the water absorption of PS and PVA is shown. The skate-shaped graphs demonstrate that an increase in kaolin content resulted in a decrease in water absorption. Specifically, Expt 11, Expt 13, and Expt 14 exhibited maximal absorption with 1%, 1%, and 0% wt of kaolin, respectively, while Expt 1, Expt 2, and Expt 3 demonstrated minimal water absorption with 10%, 13%, and 10% wt of kaolin, respectively (as shown in Table 2). These findings are further supported by the predicted absorption values depicted in Figure 2C, which ranged from 5% to 58.68%. This finding is consistent with the results reported by Kaewtatip et al. [43], who demonstrated that kaolin functions as an obstacle within the matrix. These observations collectively indicate that the presence of kaolin does indeed inhibit water absorption.

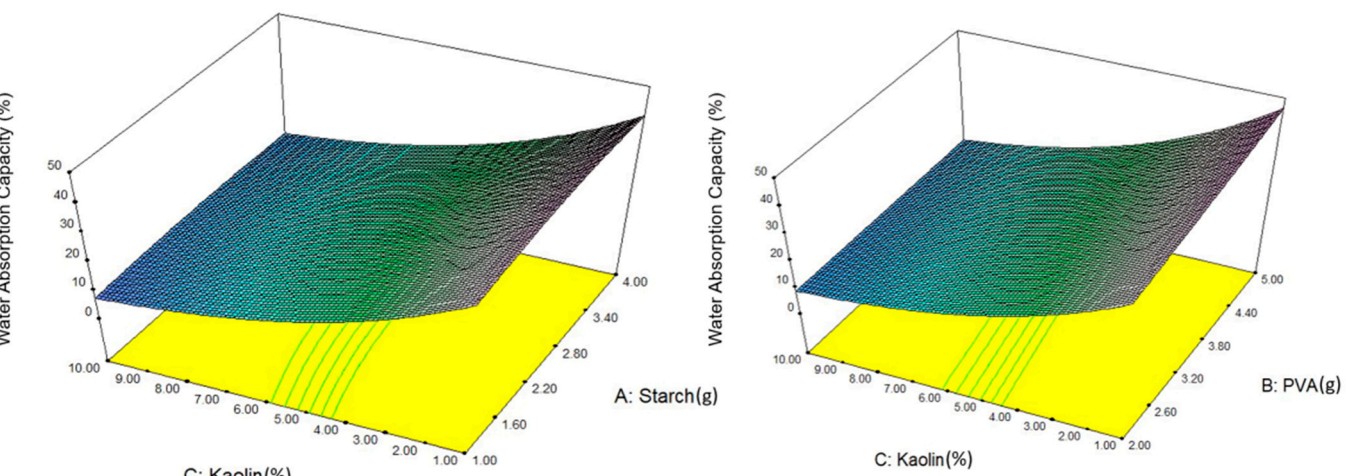

**Figure 4.** Three-dimensional response surface methodology (RSM) graphs showing the impact of the interaction of kaolin with starch and PVA on the water absorption capacity (%) of the prepared films.

As starch and PVA are hydrophilic and easily absorb water [44], the introduction of kaolin helped to decrease the water absorption capacity and increase the reliability of the films for materials with water content. This is due to kaolin's ability to synergistically interact with the matrix components and reduce the water absorption capacity. Additionally, the proper dispersion of kaolin into the polymer matrix and its good compatibility with the other components further improve the properties of the films [45].

### 4.4. Effect on Moisture Content

Moisture content is a crucial factor to consider, as it can significantly influence the physical and mechanical properties of composite films. Figure 5 displays the impact of the moisture content on the characteristics of composite films made from PVA, starch, and kaolin. The moisture content data range from 2 to 50% and are represented in Figure 2C. Table 2 reveals that the moisture content of Expt 2 with 13.07 wt% of kaolin is 4%, while Expt 13 with 1 wt% of kaolin has a moisture content of 40%. The films produced in Expt 7 and Expt 16 exhibited the lowest moisture contents. Overall, the results showed that the moisture content of synthesized composite films decreased with increasing kaolin concentration up to a certain point.

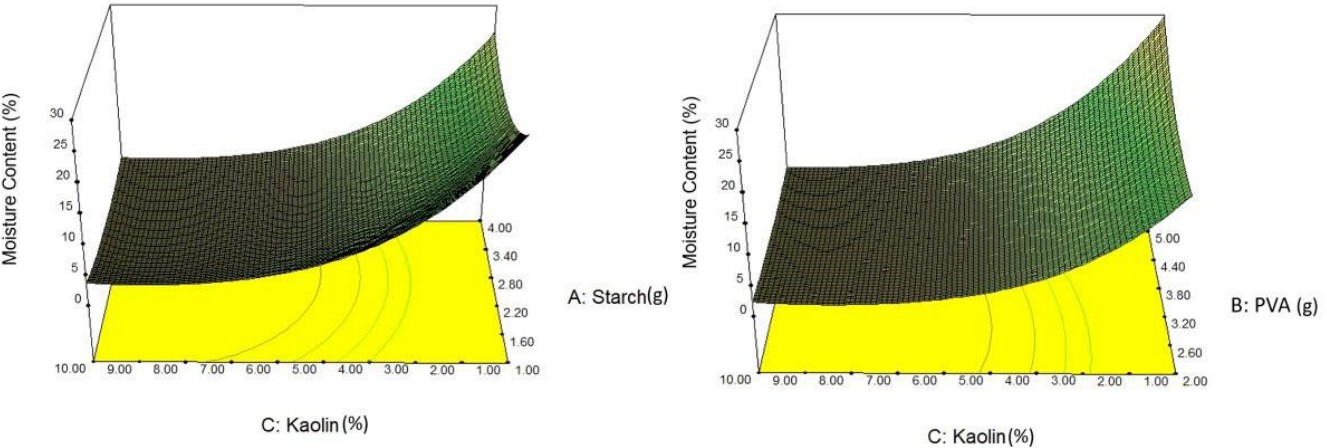

**Figure 5.** Three-dimensional response surface methodology (RSM) graphs showing the impact of the interaction of kaolin with starch and PVA on the moisture content (%) of the prepared films.

The reason behind the observed phenomenon is that kaolin, when properly dispersed in the polymer matrix, creates a barrier against water diffusion due to its non-swelling and water-resistant properties. Additionally, the presence of aluminum silicate, along with small amounts of iron, magnesium, and calcium oxides, further improves the barrier properties of the films, reducing their permeability to moisture. This can be easily observed in Figure 4, where the 3D slide-shaped graphs of moisture content demonstrate that increasing the kaolin content results in a decrease in the rate of moisture uptake, making the films more stable in humid conditions. These findings are consistent with other studies [46–48].

Furthermore, the compatibility of kaolin with starch and PVA enhances the interfacial adhesion and reduces the tendency of the films to delaminate or disintegrate upon exposure to moisture [49]. This is due to the strong chemical bonds formed between the hydroxyl groups of the PVA and starch and the surface of the kaolin particles. These chemical bonds provide excellent compatibility and stability to composite films.

### 4.5. Effect on Degradability

The results of the degradability test conducted in sandy loam soil are presented in Table 2, showing that Expt 16 exhibited the highest degradability (45%) with 5.5 wt% of kaolin, while Expt 6 had the lowest degradability (16%) with 1 wt% of kaolin. Additionally, Figure 2C displays the predicted data of degradability, ranging from a maximum of 48% with 5.5 wt% kaolin to a minimum of 15.5% with 1 wt% of kaolin. Figure 6, representing curve-shaped graphs, indicates an increased concentration of kaolin, while the low PS and high PVA levels of the matrix increased the degradability of the films. These results suggest that the addition of kaolin in the PVA/PS matrix at a certain level enhances the breakdown characteristics of the films. This may be attributed to the compact structure of kaolin, which contains 1:1 uncharged dioctahedral layers in its composition, as well as the strong intermolecular OH interactions between PVA/PS and kaolin. These findings are consistent with previous studies [50–52].

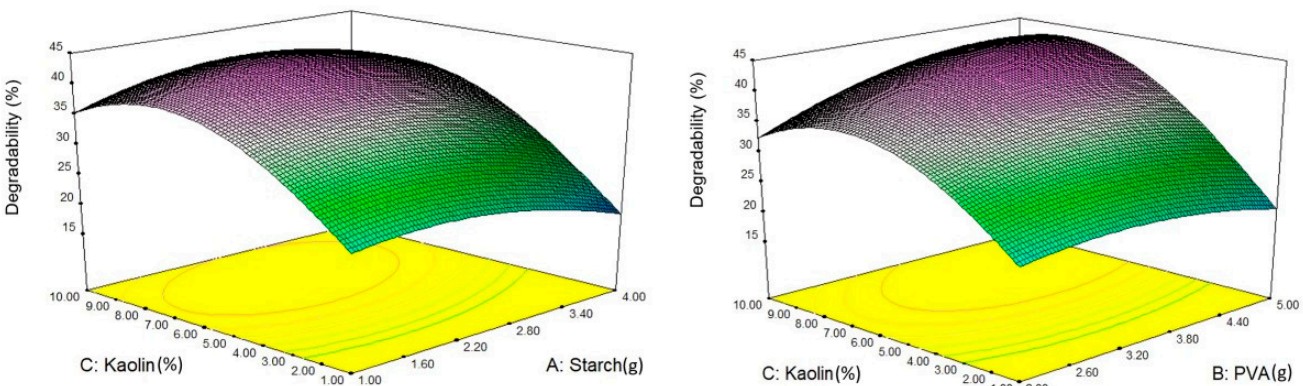

**Figure 6.** Three-dimensional response surface methodology (RSM) graphs showing the impact of the interaction of kaolin with starch and PVA on the degradability (%) of the prepared films.

### 4.6. Optimization and Validation

It was observed that there is potential for all developed PVA/PS kaolin-based composite films as packaging materials. For this application, it is necessary to identify the optimum ratios of PVA/PS kaolin to obtain the best film. By using RSM software, optimization was carried out. The optimization of tensile strength and moisture content was carried out at their maximum and minimum values, while the remaining three responses were kept constant.

Numerical optimization was conducted by setting the tensile strength and moisture content at their maximum and minimum possible values, while the other three responses were kept within their ranges. The optimal conditions were 5.5 wt% of kaolin, 2.5 g of starch, and 3.5 g of PVA. The predicted responses were 26.5 MPa, 96%, 21%, 3%, and 48% for tensile strength, elongation at break, water absorption capacity, moisture content, and degradability, respectively (Figure 2).

Validation experiments were performed using the optimal conditions three times, and the average values obtained for tensile strength, elongation at break, water absorption capacity, moisture content, and degradability were 20.5 MPa, 94%, 22%, 3%, and 45%, respectively (Table 2). The error between the predicted and experimental values was less than 10%, indicating a good agreement between the two. The optimized mechanical properties of the composite film were found to be comparable to those of conventional petroleum-based packaging films, with the added advantage of significant degradability.

The results for the optimal sample were consistent with those reported by other research groups using similar compositions. For instance, one study reported tensile strength, elongation at break, and WVP values of 35 MPa, 50%, and $5.6 \times 10^{-8}$ g/s/m, respectively, for starch-based composites with a different plasticizer and PVA [53]. In another study, the tensile strength, elongation at break, and WVP were reported as 10 MPa, 40%, and $38 \times 10^{-8}$ g/s/m, respectively, for a blend of biopolymers and PVA [54]. A third study reported tensile strength and WVP values of 100 MPa and $0.21 \times 10^{-8}$ g/s/m, respectively, for a composite film containing a natural polymer, PVA, and a biodegradable plasticizer [55]. Lastly, a study found that a composite film made from a starch-based material, PVA, and a nanofiller exhibited tensile strength, elongation at break, and WVP values of 25 MPa, 185%, and $5.2 \times 10^{-8}$ g/s/m, respectively [56].

### 4.7. Structural and Morphological Analysis

Figure 7 presents the FT-IR spectra of kaolin, the matrix (PVA/starch), and the optimum composite film. The FTIR spectrum of kaolin presents a peak at 3587 cm$^{-1}$, which is associated with the hydroxyl group's stretching vibrations, while the peak at 1654 cm$^{-1}$ corresponds to bending vibrations of the similar group adsorbed on the surface of kaolin. The stretching vibrations of Si-O bonds represent the peaks at 962 cm$^{-1}$ and 804 cm$^{-1}$, characterizing the presence of silicon and oxygen atoms bonded together in a mineral

lattice. The peak at 848 cm$^{-1}$ can be attributed to the bending vibrations of aluminum and oxygen bonds. The bending vibrations of aluminosilicate (Si-O-Al) are the most intensive band shown at 671 cm$^{-1}$, showing the presence of silicon, oxygen, and aluminum atoms bonded together in the kaolin sample. This bond is central, as it confirms the unique composition of kaolin and contributes to the stability and structure of kaolin. An in-plane stretching vibration is shown at 430 cm$^{-1}$ and is related to the siloxane bond [57,58].

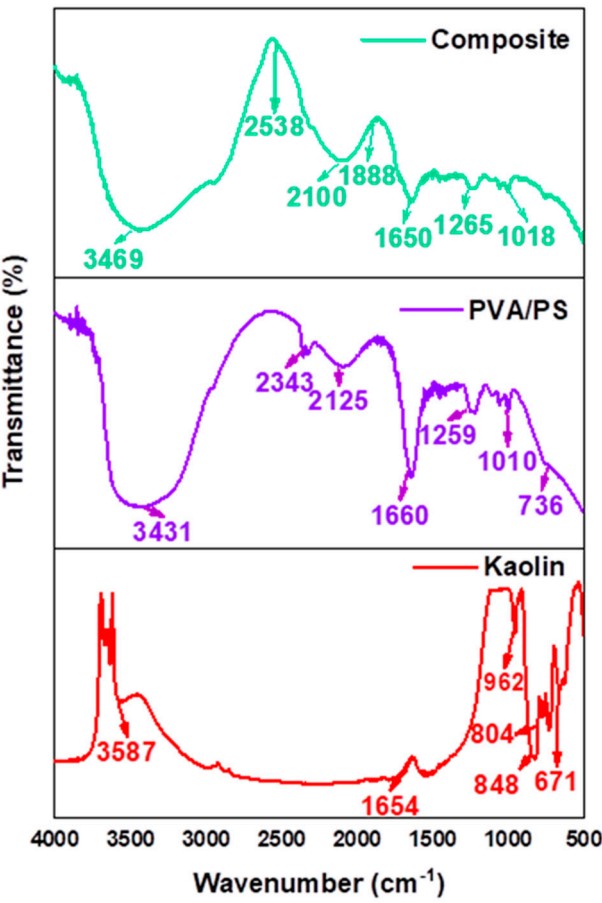

**Figure 7.** FTIR spectra of kaolin, PVA/PS, and composite.

In Figure 7 and Table 5, the FT-IR spectra of the matrix (PVA/PS) can be seen, presenting hydroxyl group stretching vibrations at 3431 cm$^{-1}$ and carbonyl group (C=O) stretching vibrations at 2343 cm$^{-1}$. The peak at 2125 cm$^{-1}$ corresponds to the stretching vibration of the C-O bond in the PVA/PS matrix. It indicates the presence of glycosidic bonds, which are formed between the glucose units in starch molecules. Moreover, it is the characteristic feature of the polymer's backbone [59]. Additionally, the stretching vibration of the carbon–carbon double bond (C=C) at 1660 cm$^{-1}$ indicates the presence of unsaturated carbon–carbon bonds. This double bond contributes to the flexibility and reactivity of the composite, allowing for potential crosslinking or polymerization reactions. An oxygen atom bridging two carbon atoms through a covalent bond (C-O-C) was observed at 1010 cm$^{-1}$, which is indicative of the structure of the polymeric carbohydrate chain. Moreover, C-H at a wavelength of 736 cm$^{-1}$ corresponds to the stretching vibration of carbon–hydrogen (C-H) bonds in the PVA/PS matrix, indicating the presence of organic groups containing carbon and hydrogen atoms.

**Table 5.** FTIR frequency range and the functional groups present in samples.

| Samples | Functional Groups | Peaks Observed (cm$^{-1}$) | References |
|---|---|---|---|
| Kaolin | O-H | 3587 | [45–48] |
| | O-H | 1654 | |
| | Si-O | 962 | |
| | Si-O | 804 | |
| | Al-O | 848 | |
| | Si-O-Al | 671 | |
| PVA/PS | O-H | 3431 | [25,32,47,49] |
| | C=O | 2343 | |
| | C-O | 2125 | |
| | C=C | 1660 | |
| | O-H | 1259 | |
| | C-O-C | 1010 | |
| | C-H | 736 | |
| Composite | O-H | 3469 | [46–52] |
| | C=O | 2538 | |
| | Si-O | 2100 | |
| | C=C | 1888 | |
| | H-O-H | 1650 | |
| | C-O | 1265 | |
| | Si-O-Si | 1018 | |

In contrast, upon the incorporation of kaolin into the PVA/PS matrix as a strong reinforcing agent, significant shifts in the peaks of the optimal composite can be observed. As shown in Table 5, the peak at 3469 cm$^{-1}$ corresponds to the stretching vibrations of hydroxyl groups (O-H) present in both PVA/PS (3431 cm$^{-1}$) and kaolin (3787 cm$^{-1}$), indicating the presence of strong hydrogen bonding within the composite. Notably, new peaks emerge at 2538 cm$^{-1}$ for carbonyl stretching vibrations and at 1888 cm$^{-1}$ for carbon double bonds, demonstrating the enhanced stability and functionality of the optimal composite. Furthermore, the presence of the peak at 1265 cm$^{-1}$, associated with the C-O carbonyl group, confirms the successful integration of PVA and PS in the kaolin matrix.

Figure 7a illustrates that kaolin exhibits an Si-O-Al bond peak at 671 cm$^{-1}$ whereas, in the spectra of the optimal composite, this bond shifts to Si-O-Si. This shift occurs because of chemical and physical interactions that take place when the optimal concentration of kaolin is incorporated into the PVA/PS matrix. The interactions between the hydroxyl groups of PVA and PS with the aluminum ions in kaolin lead to the detachment of aluminum and the subsequent attachment of silicon, resulting in the prevalence of the Si-O-Si bond in the optimal composite. This shift from Si-O-Al to Si-O-Si confirms the successful integration of kaolin into the PVA/PS matrix and highlights the chemical compatibility and potential crosslinking or interactions between the components, ultimately contributing to the overall structure and properties of the composite material. These findings have previously been observed by other researchers [60,61].

Figure 8 illustrates the TGA curves of kaolin, the matrix (PVA/potato starch), and the optimum composite film. All the samples underwent three-step degradation, and the results are tabulated in Table 6. Two distinct peaks are observed in the TGA curve of kaolin, one at around 110 °C and the other at 448 °C. The initial peak at 110 °C indicates a mass loss of 6.7%, attributed to the removal of water molecules. The significant mass loss observed

at 450 °C (9.9%) can be attributed to dehydroxylation, which involves the diffusion and reorganization of the hydroxyl groups within the layers of kaolin [60,61].

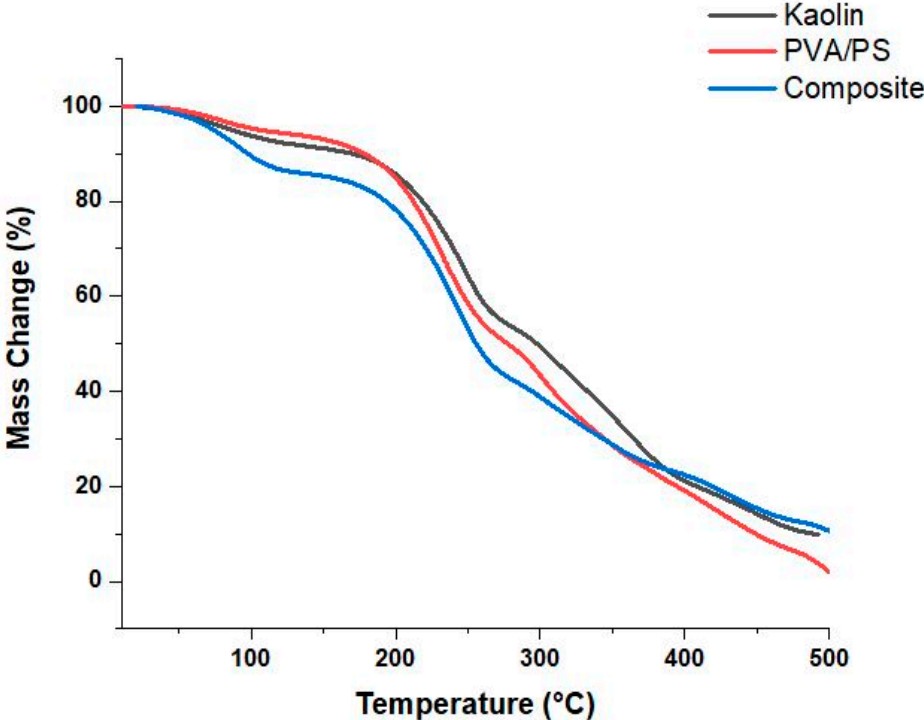

**Figure 8.** TGA curves of kaolin, PVA/PS, and Composite.

**Table 6.** Data of TGA for samples of kaolin, PVA/PS, and composite.

| Sample | 1st Peak (°C) | 2nd Peak (°C) | 3rd Peak (°C) | Residue Left (%) |
|---|---|---|---|---|
| Kaolin | 110 | 250 | 450 | 9.9 |
| PVA/PS | 150 | 268 | 475 | 5.5 |
| Composite | 180 | 275 | 499 | 18.68 |

The first and second degradation peaks appeared at 150 °C and 268 °C, respectively, for PVA/PS. The third degradation peak also remained almost unchanged when compared to raw kaolin. In the obtained optimum film, the initial degradation peak shifted towards higher temperatures, from 150 °C in PVA/PS to 180 °C, and the third peak from 475 °C to 499 °C, with residual masses ranging from 5.5 to 18.68%, respectively. This shows that the elevated thermal stability of the composite is a combination of both the kaolin and matrix components. Similar trends were also reported in many other studies [19,26,27,51].

Using SEM, the surface morphology of the obtained optimum film was examined at three different magnifications (Figure 9). At 1 kx magnification (Figure 9a), the surface of the composite film appears smooth and homogeneous, with no visible cracks or defects. The surface morphology shows a relatively uniform distribution of kaolin particles, which appear as white spots in the matrix of PVA and starch. At 10 kx magnification (Figure 9b), the kaolin particles become more visible and appear as irregularly shaped clusters or aggregates, distributed uniformly throughout the PVA/starch matrix. The boundaries between the kaolin particles and the matrix are well defined, indicating good interfacial adhesion between the two phases.

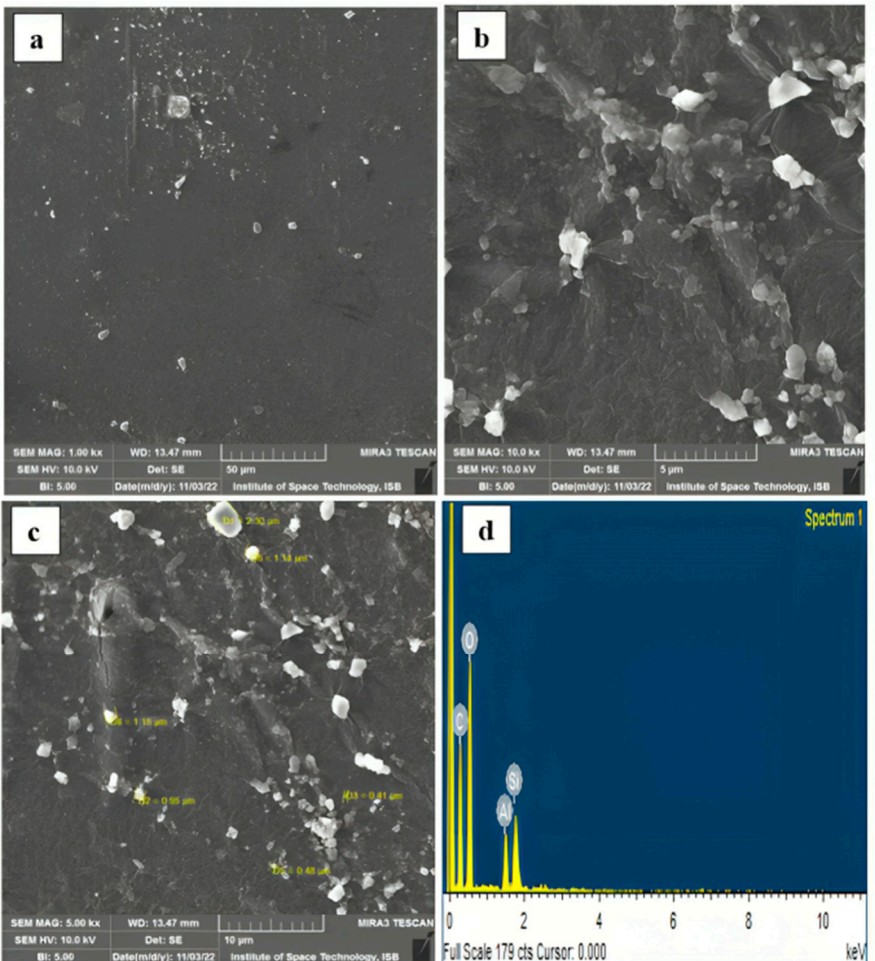

**Figure 9.** SEM images of obtained optimum composite at (**a**) 1.0 kx, (**b**) 10.0 kx, and (**c**) 5.00 kx magnifications and (**d**) EDX of optimum film.

At 50 kx magnification (Figure 9c), the surface of the composite film shows a high degree of roughness, with several small-scale irregularities and surface features. The kaolin particles are visible and appear as small, irregularly shaped particles with rough surfaces ranging from 0.48 μm to 2.30 μm. The PVA/potato starch matrix surrounding the kaolin particles also shows some roughness, which may be due to the formation of interfacial layers or bonding between the kaolin particles and the matrix. The EDX elemental analysis shown in Figure 9d was conducted to confirm the composition of the obtained optimum composite, revealing the following percentages: carbon, 35.50%; oxygen, 49.57%; aluminum, 6.38%; and silica, 8.55%. The results suggest that the composite has a higher amount of silicon compared to aluminum. This is attributed to the inherent characteristics of kaolin, which is rich in silica and contains a lower percentage of aluminum. These findings correlate with other studies [50,53,55,58].

## 5. Conclusions

In the present investigation, starch/PVA/kaolin composite films were prepared and then optimized by Design Expert software. The set variables in the experimental design were the amount of potato starch, PVA, and kaolin, and the responses were mechanical properties, water absorption capacity, moisture content, and degradability. The main conclusions are as follows:

- The mechanical analysis revealed that the incorporation of kaolin into the PS and PVA matrix has a significant impact. Obtaining the optimum concentration for all components is crucial for achieving a balance between TS and EAB. The optimal level

was found to be 5.5 wt% of kaolin, 3.5 g of PVA, and 2.50 g of potato, resulting in good mechanical properties. If the concentrations of PVA and PS fluctuate in conjunction with kaolin, a significant difference in tensile strength (TS) and elongation at break (EAB) is observed.

- The fluctuating concentration of kaolin exhibited a trend of both increase and decrease in the water absorption capacity of the composite films. In this way, the maximum water absorption capacity was observed in Expt 14 of 51% with 0.0 wt% of kaolin, and Expt 2 exhibited a lower water absorption capacity of 6% with 13.07 wt% of kaolin. The absence of kaolin in the PVA/PS matrix demonstrated an improved water absorption capacity because both PVA and PS are hydrophilic in nature and their OH groups align with polymer chains. The higher concentration of kaolin in the second case increased its swelling properties, creating a barrier against water and making the film brittle in nature. This indicates the crucial role of the optimum value of kaolin in achieving a balanced and improved water absorption capacity.
- A similar trend in water absorption properties was observed in the properties of moisture content and degradability. A higher moisture content and degradability were observed in Expt 13 and Expt 16, while Expt 16 and Expt 6 showed lower moisture content and degradability, respectively.
- The FTIR results revealed Si-O bonds and Si-O-Al at 962 $cm^{-1}$ and 858 $cm^{-1}$, respectively, confirming the unique 1:1 composition of kaolin. Furthermore, the C=C stretching vibrations in the PVA and PS polymer matrix were observed at 1660 $cm^{-1}$, representing polymerization reactions and reactivity within the matrix. A significant bond shift was observed when incorporating kaolin into the PVA–PS matrix, where the Si-O-Al bond peak replaced the Si-O-Si peak. This indicates the successful integration of kaolin into the composite, highlighting the coherence between its components.
- The SEM-EDX analysis performed on the obtained optimum composite film highlighted a uniform and smooth surface at 1 kx magnification, indicating strong interactions within the components of the matrix and kaolin. The EDX analysis confirmed the major elemental groups representing the proper formation of the composite, with clear percentages of carbon (35.50%), oxygen (49.57%), aluminum (6.38%), and silica (8.55%). The TGA analysis revealed the excellent thermal stability of the optimum composite film, as indicated by the maximum residual of 18.68%.
- An ANOVA and 3D graphs were employed to analyze the relationship between the parameters and responses. The coefficient of determination (R2) was utilized to assess the goodness of fit of the model to the data. Based on the model statistical analysis, all the models were significant at a *p*-value lower than 0.05, the F-statistic was not significant, Adj $R^2$ values were close to Pred $R^2$ values, and the CV% values were also below 10%. This suggests that there is a strong correlation between the process factors and the relevant responses, and all the models are reproducible.

The study results indicated that the inclusion of kaolin led to enhanced mechanical and thermal properties of the composite film matrix. These findings suggest that the PVA/PS-based kaolin composite material has good potential for packaging applications.

**Author Contributions:** Conceptualization, M.Q.; methodology, M.Q.; software, M.Q.; validation, M.Q., N.T., A.B. and S.A.; formal analysis, M.Q.; investigation, M.Q.; resources, N.T., U.R. and A.A.A.M.; data curation, M.Q.; writing—original draft preparation, M.Q.; writing—review and editing, M.Q., N.T., S.A. and A.B.; visualization, M.Q.; supervision, N.T. and U.R.; project administration, N.T., U.R. and A.A.A.M.; funding acquisition, N.T. and A.A.A.M. All authors have read and agreed to the published version of the manuscript.

**Funding:** This research was funded by Fatima Jinnah Women University, Rawalpindi, Pakistan (Grant No. ORIC-FJWU/2020-21/1008) and funded by the Researchers Supporting Project Number (Grant No. RSPD2023R766) King Saud University, Riyadh, Saudi Arabia.

**Data Availability Statement:** Data are contained within the article.

**Acknowledgments:** The authors would like to acknowledge Institute of Space Technology, Islamabad, Pakistan for SEM analysis and Lahore College for Women University, Lahore, Pakistan for TGA analysis. The authors thank Saqib Jabbar (NARC, Islamabad) for Design Expert Software assistance.

**Conflicts of Interest:** The authors declare no conflicts of interest.

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
