# Peer review of "Kaolin–Polyvinyl Alcohol–Potato Starch Composite Films for Environmentally Friendly Packaging: Optimization and Characterization"

_jcs, doi:10.3390/jcs8010029_

Round 1

Reviewer 1 Report

Comments and Suggestions for Authors

This paper introduces a method for the production of biode-13 gradable composite films by combining kaolin, polyvinyl alcohol (PVA), and potato starch (PS) 14 using a solvent casting technique. The sample had tensile strength of 21 26.5 MPa, an elongation at break of 96%, a water absorption capacity of 21%, a moisture content of 22 3%, and a remarkable degradability of 48%. The properties were good, but some problems should be addressed and major revision is recommended.

1. In section 2.1, how did the authors ensure the thickness uniformity of the fabricated films? The film fabrication process should be explained with more details.

2. How did the authors carry out the tensile test as the film was thin? The clamp setting of the mechanical experiments should be given.

3. What is the mechanism of the water absorption for the proposed film? The authors should explain the water absorption properties.

4. In Fig. 3, the X and Y coordinate did not have a unit. Are they the concentration of the components? The authors should revise the figure and the corresponding explanation.

5. Some important composites with good mechanical and electromagnetic properties should be cited for better introduction as follows

(1) https://doi.org/10.1016/j.compositesb.2023.110737

(2) https://doi.org/10.1016/j.cej.2022.140247

Comments on the Quality of English Language

The language should be improved.

Author Response

Reviewer # 1

  1. In section 2.1, how did the authors ensure the thickness uniformity of the fabricated films? The film fabrication process should be explained with more details.

See Page 4, Line 170-175 (“Section 2.1. Preparation of films”)

According to the respected reviewer's suggestion, the authors ensured the uniformity/thickness of all films. In the relevant section, it is explained that a film casting knife was used to ensure equal spreading of films and to confirm uniform thickness, a digital micrometer was employed. Readings were taken from five points on each film—four sides and one center—and the average was calculated to ensure uniformity.Top of Form

  1. How did the authors carry out the tensile test as the film was thin? The clamp setting of the mechanical experiments should be given.

This has been added (see page 5,6 – Lines 201-2016 “Mechanical Analysis Section 3.4” )

In accordance with the reviewer's suggestion, changes have been incorporated into the relevant section. The reviewer noted that since the films were thin, details about the mechanical analysis, including clamp settings, should be provided. The mechanical analysis was conducted using the WAW-500 E computerized hydraulic machine, which is designed for both round and flat samples. For flat samples, the machine allows a maximum thickness range of 0-30mm, which falls within the range of our films.

For the convenience of the reviewer, the information recommended to be incorporated into the relevant section has been provided below in square brackets. Additionally, these changes have been included in the paper with track changes.

[The tensile strength (TS, MPa) and elongation at break (EAB, %) of the films were measured by following ASTM D882-12 using the WAW-500E computerized hydraulic universal tensile testing machine  [32,33]. This machine provides a maximum load of 500kN with ≤±1% load and deformation accuracy. The machine has two important zones for testing film: the upper zone, which is fixed and provides tension, and the lower zone, which is movable for compression, stretching, and bending of the sample. The machine's clamps are designed to accommodate both round and flat samples. However, in this study, flat clamps with a thickness range of 0-30 mm were used for the fabricated films. The machine offers a maximum distance of 750 mm between the two tensile grips. In accordance with the fabricated films of this study, the initial adjustment was set at 40 mm, and before testing, all films were cut into strips (100 mm × 40 mm) and left to stabilize at 24 â—¦C and 50% Relative Humidity for three days [6]. The ends of the films were mounted vertically between the clamps, and the extensional speed was set to 0.2 kn/sec. A photoelectric encoder, a sensing device installed in the machine, transmitted the movements of the samples outside as an electrical signal. Tensile data were collected using software, and to ensure accuracy, each experiment was repeated three times].

  1. What is the mechanism of the water absorption for the proposed film? The authors should explain the water absorption properties.

See Page 6, Line 219-234 “Section3.5 Water Absorption”

As per the reviewer's suggestion, the relevant section has been updated to include information on the water absorption mechanism and the chemistry behind it concerning major components of composite films, such as kaolin and the PVA/PS matrix, along with their properties.

Below, the entire section has been attached within square brackets, involving the incorporated relevant information.

[Water absorption capacity is considered a crucial property for evaluating the performance of bio-based films in diverse applications. It deals with the ability of bioplastic films to take in and retain water under explicit conditions. The water potential of the films is affected by factors such as the presence of hydrophilic functional groups in the polymer structure, the degree of crystallinity, and the porosity of the film. In this investigation, the water absorption potential of the films composed of kaolin and the matrix PVA/PS were assessed in triplicate using a method described by Shen et al.,[34] with minor adaptations. The hydrophilic nature of PVA/PS arises from the presence of hydroxyl (-OH) groups along their polymer chains. These OH groups readily interact with water molecules through H-bonding, facilitating the absorption of water into the matrix. In addition to that, understanding the chemistry of kaolin with respect to water absorption is crucial, especially when incorporating it into matrices. Kaolin possesses a porous and layered crystal structure with OH groups on its surface. When it interacts with PVA and PS, it enhances the overall adhesion, stability, and compatibility between the components, contributing to improved water absorption properties of the film.

This test was performed by submerging dry 50 x 50 mm films in 100 ml distilled water at room temperature for 24 hours, and the non-soluble film was examined for any dimensional changes after gently dabbing the surface with a Whatman filter paper. The water absorption capacity was calculated according to the equation WA (%) = [(Ww-Wd)/Wd]x100, where Ww is the weight of the wet film after sinking, and Wd is the weight of the dry film before sinking ]. Top of Form

  1. In Fig. 3, the X and Y coordinate did not have a unit. Are they the concentration of the components? The authors should revise the figure and the corresponding explanation.

See pages 11, 12, Lines-313-331 (Section 4.2 “Effect on Mechanical Properties”)

Yes, the reviewer has rightly pointed out that the units for the X1 and X2 coordinates on the 3D graphs of RSM were not specified. These coordinates represent concentrations of the components. The units have now been added to the graphs, and other relevant information related to Figure 3 has also been appropriately modified.

  1. Some important composites with good mechanical and electromagnetic properties should be cited for better introduction as follows

(1) https://doi.org/10.1016/j.compositesb.2023.110737

(2) https://doi.org/10.1016/j.cej.2022.140247

Yes, some additional information on some other composites has been included in the sections below from the latest and suggested research papers and proper references have been incorporated.

See page 2, Lines 95-99  (Section 1 “Introduction”)

See page 11, Lines 307-310  (Section 4.2 “Effect on Mechanical Properties”)

  1. The language should be improved.

The paper has been extensively revised for language with the help of native speaker and additionally rechecked by Grammarly software.

Reviewer 2 Report

Comments and Suggestions for Authors

Regarding the study “Kaolin- Polyvinyl Alcohol -potato Starch Composite Films for Environmentally Friendly Packaging: Optimization and Characterization”, I felt confident that the authors conducted a meticulous and comprehensive investigation. However, I also observed that certain essential aspects were inadequately described. In the following section, I will elaborate on my specific concerns in more detail.

The introduction references statistics related to the plastic sector. To enhance the credibility and provide readers with the latest information, it would be advantageous to include not only the Agarwal citation but also the sources and years of other pertinent statistics. This ensures accessibility to the most recent and reliable data for readers.

To ensure the production of a thorough and balanced research article, I kindly request a revision of the "Conclusion" section. Specifically, I propose incorporating additional information pertaining to the samples (excluding Exp ..insert number..). Moreover, clarity regarding the optimization timeline and the specific sample subjected to FTIR, SEM-EDX analyses would enhance the overall comprehension of this section.

Author Response

Reviewer # 2

Regarding the study “Kaolin- Polyvinyl Alcohol -potato Starch Composite Films for Environmentally Friendly Packaging: Optimization and Characterization”, I felt confident that the authors conducted a meticulous and comprehensive investigation. However, I also observed that certain essential aspects were inadequately described. In the following section, I will elaborate on my specific concerns in more detail.

Author’s would like to thank the respected reviewer for positive comments and suggestions on our paper. We fully tried to incorporate all suggestions by respected reviewer as follows.

The introduction references statistics related to the plastic sector. To enhance the credibility and provide readers with the latest information, it would be advantageous to include not only the Agarwal citation but also the sources and years of other pertinent statistics. This ensures accessibility to the most recent and reliable data for readers.

See Pages 1,2 - Lines 43-63 (Section 1 “Introduction”)

As per the reviewer's suggestion to enhance the credibility of the paper, additional statistics from the recent data from the plastic sector should be incorporated. The required information is enclosed in square brackets for the reviewer's convenience, and it is also mentioned in the paper with track changes.

[The packaging sector, encompassing items such as plastic bags, plastic bottles, and product wrappings, contributes to 30% of the total plastic consumption. Similarly, the building and construction sector, involving plastic pipes and vinyl coverings, accounts for 17%, while the transportation sector, including glazing, interior wall panels, partitions, and headliners, constitutes 14% [3]. The most used synthetic plastics across these sectors are low- and high-density polyethylene (LDPE, HDPE), polypropylene (PP), polyvinyl chloride (PVC), polystyrene (PS), and polyethylene terephthalate (PET). Collectively, these plastics make up approximately 90% of the total plastic production [5]. The major raw materials used for packaging applications are polyethylene and polypropylene, which contribute significantly to the country's global economy [2]. Despite their low cost and good mechanical and thermal properties, these materials are synthesized from crude oil byproducts, making them the main culprit for global warming [4].
In addition to this, plastic pollution is causing the deaths of aquatic animals, loss of habitats, and the pollution of landfills. According to previous studies, the arctic ice, once considered a pure and virgin environment, is now found to be contaminated with microplastics, with up to 10 thousand particles per liter of snow observed [8,10]. According to Lau et al.,[11] if the current rate of plastic usage continues, it is projected that by 2050, 12 billion tons of plastic will accumulate in landfills and the environment. The consequence of this would be a depletion of 20% of our world's natural resources].

Top of Form

To ensure the production of a thorough and balanced research article, I kindly request a revision of the "Conclusion" section. Specifically, I propose incorporating additional information pertaining to the samples (excluding Exp ..insert number..). Moreover, clarity regarding the optimization timeline and the specific sample subjected to FTIR, SEM-EDX analyses would enhance the overall comprehension of this section.

See pages 21, 22- Lines 585-637, (Section 5 “Conclusion”)

As per the reviewer's second comment, it was recommended to revise the conclusion section. Accordingly, the conclusion section has undergone a complete revision. Apart from experiment numbers, trends related to performed experiments w.r.t specific analysis, as suggested by the reviewer, have been incorporated. Further, possible reasoning with respect to FTIR and bonding information of the composite film, specifically related to kaolin, has been added. Elaboration of information from SEM and EDX analyses has also been extended. The complete revised conclusion section is attached below in square brackets, and the entire conclusion is available with track changes.

[In the present investigation, starch/PVA/kaolin composite films were prepared and then optimized by Design Expert Software. The set variables in the experimental design were the amount of potato starch, PVA, and kaolin and the responses were mechanical properties, water absorption capacity, moisture content, and degradability.  The main conclusions include:

  • The mechanical analysis revealed that the incorporation of kaolin into the PS and PVA matrix has a significant impact. Obtaining the optimum concentration for all components is crucial for achieving a balance between TS and EAB. The optimal level was found to be 5.5 wt% kaolin, 3.5 g PVA, and 2.50 g potato, resulting in good mechanical properties. If the concentrations of PVA and PS fluctuate in conjunction with kaolin, a significant difference in tensile strength (TS) and elongation at break (EAB) is observed.
  • The fluctuating concentration of kaolin exhibited a trend of both increase and decrease in the water absorption capacity of the composite films. In this way, the maximum water absorption capacity was observed in Expt-14 of 51 % with 0.0 wt% kaolin and Expt-2- exhibited a lower water absorption capacity of 6% with 13.07 wt% kaolin.
    The absence of kaolin in the PVA/PS matrix demonstrated an improved water absorption capacity because both PVA and PS are hydrophilic in nature, and their OH groups align with polymer chains. The higher concentration of kaolin in the second case increased its swelling properties, creating a barrier against water and making the film brittle in nature. This indicates the crucial role of the optimum value of kaolin for achieving a balanced and improved water absorption capacity.
  • A similar trend in water absorption properties was observed in the properties of moisture content and degradability. The higher moisture content and degradability were observed in Expt-13 and Expt-16, while Expt-16 and Expt-6 showed lower moisture content and degradability, respectively.
  • The FTIR results revealed Si-O bonds and Si-O-Al at 962 cm-1 and 858 cm-1, respectively, confirming the unique 1:1 composition of kaolin. Furthermore, the C=C stretching vibrations in the PVA and PS polymer matrix were observed at 1660 cm-1, representing polymerization reactions and reactivity within the matrix. A significant bond shift was observed when incorporating kaolin into the PVA-PS matrix, where the Si-O-Al bond peak replaced the Si-O-Si. This indicates the successful integration of kaolin into the composite, highlighting the coherence between its components.
  • SEM-EDX analysis performed on the obtained optimum composite film highlighted a uniform and smooth surface at 1kx magnification, indicating strong interactions within the components of the matrix and kaolin. The EDX analysis confirmed the major elemental groups representing the proper formation of the composite, with clear percentages of Carbon (35.50%), Oxygen (49.57%), Aluminum (6.38%), and Silica (8.55%).TGA analysis revealed the excellent thermal stability of optimum composite film as indicated by the maximum residual of 18.68%. 
  • ANOVA and 3D graphs were employed to analyze the relationship between parameters and responses. The coefficient of determination (R2) was utilized to assess the goodness of fit of the model to the data. Based on the model statistical analysis, all the models were significant at a P-value lower than 0.05, the F-statistic was not significant, Adj R2 values were close to Pred R2 values and CV% were also below 10%. This suggests that there is a strong correlation between process factors and relevant responses and all models are reproducible. 

The study results indicated that the inclusion of kaolin led to enhanced mechanical and thermal properties of the composite film matrix. These findings suggest that the PVA/PS-based kaolin composite material has good potential for packaging applications ].

Round 2

Reviewer 1 Report

Comments and Suggestions for Authors

Some important composites with good mechanical and electromagnetic properties should be cited for better introduction as follows,but the authors did not add the two following references in the manuscript. Please revise the references.

(1) https://doi.org/10.1016/j.compositesb.2023.110737

(2) https://doi.org/10.1016/j.cej.2022.140247

Comments on the Quality of English Language

The English should be improved.

Reviewer 2 Report

Comments and Suggestions for Authors

Accept in present form